# Inter-species variation in monovalent anion substrate selectivity and inhibitor sensitivity in the sodium iodide symporter (NIS)

**Susanna C. Concilio[1,2], Hristina R. Zhekova[3], Sergei Y. Noskov[3], Stephen J. Russell[1] ***

**1** Department of Molecular Medicine, Mayo Clinic, Rochester, Minnesota, United States of America, **2** Mayo Clinic Graduate School of Biomedical Sciences, Mayo Clinic, Rochester, Minnesota, United States of America, **3** Centre for Molecular Simulation, Department of Biological Sciences, University of Calgary, Calgary, Alberta, Canada

\* sjr@mayo.edu

**Data Availability Statement:** We have deposited our data into Open Science Framework (https://osf. io) with the DOI: 10.17605/OSF.IO/3YG7J.

## Abstract

The sodium iodide symporter (NIS) transports iodide, which is necessary for thyroid hormone production. NIS also transports other monovalent anions such as tetrafluoroborate ($BF_4^-$), pertechnetate ($TcO_4^-$), and thiocyanate ($SCN^-$), and is competitively inhibited by perchlorate ($ClO_4^-$). However, the mechanisms of substrate selectivity and inhibitor sensitivity are poorly understood. Here, a comparative approach was taken to determine whether naturally evolved NIS proteins exhibit variability in their substrate transport properties. The NIS proteins of thirteen animal species were initially assessed, and three species from environments with differing iodide availability, freshwater species *Danio rerio* (zebrafish), saltwater species *Balaenoptera acutorostrata scammoni* (minke whale), and non-aquatic mammalian species *Homo sapiens* (human) were studied in detail. NIS genes from each of these species were lentivirally transduced into HeLa cells, which were then characterized using radioisotope uptake assays, $^{125}I^-$ competitive substrate uptake assays, and kinetic assays. Homology models of human, minke whale and zebrafish NIS were used to evaluate sequence-dependent impact on the organization of $Na^+$ and $I^-$ binding pockets. Whereas each of the three proteins that were analyzed in detail concentrated iodide to a similar degree, their sensitivity to perchlorate inhibition varied significantly: minke whale NIS was the least impacted by perchlorate inhibition ($IC_{50}$ = 4.599 μM), zebrafish NIS was highly sensitive ($IC_{50}$ = 0.081 μM), and human NIS showed intermediate sensitivity ($IC_{50}$ = 1.566 μM). Further studies with fifteen additional substrates and inhibitors revealed similar patterns of iodide uptake inhibition, though the degree of $^{125}I^-$ uptake inhibition varied with each compound. Kinetic analysis revealed whale NIS had the lowest $K_m$-I and the highest $V_{max}$-I. Conversely, zebrafish NIS had the highest $K_m$ and lowest $V_{max}$. Again, human NIS was intermediate. Molecular modeling revealed a high degree of conservation in the putative ion binding pockets of NIS proteins from different species, which suggests the residues responsible for the observed differences in substrate selectivity lie elsewhere in the protein. Ongoing studies are focusing on residues in the extracellular loops of NIS as determinants of anion specificity. These data demonstrate significant transport differences between the NIS proteins of different species, which may be influenced by the unique physiological needs of

**Funding:** This study was supported by National Institutes of Health R44 TR001191 to S.J.R. (https://www.nih.gov); National Institutes of Health T32 AI132165 01A1 to S.C.C. (https://www.nih.gov); Natural Sciences and Engineering Research Council of Canada (NSERC grant No. RGPIN-315019) (https://www.nserc-crsng.gc.ca) and Alberta Innovates Technical Futures Strategic Chair in Bio-Molecular Simulations to S.Y.N. (https://albertainnovates.ca); and Alberta Innovates Health Solution Post-Doctoral Fellowship to H.R.Z. (https://albertainnovates.ca). All calculations were performed on the West-Grid/Compute Canada clusters under Research Allocation Award to S.Y.N. (https://www.westgrid.ca). The funders had no role in study design, data collection and analysis, decision to publish, or preparation of the manuscript.

**Competing interests:** I have read the journal's policy and the authors of this manuscript have the following competing interests: S.J.R. is a cofounder and holds equity in Imanis Life Sciences, LLC. S.C.C., H.R.Z., and S.Y.N. declare no competing interests. This does not alter our adherence to PLOS ONE policies on sharing data and materials.

each organism. Our results also identify naturally-existing NIS proteins with significant variability in substrate transport kinetics and inhibitor sensitivity, which suggest that the affinity and selectivity of NIS for certain substrates can be altered for biotechnological and clinical applications. Further examination of interspecies differences may improve understanding of the substrate transport mechanism.

## Introduction

The sodium iodide symporter (NIS), encoded by the gene *SLC5A5* in humans, mediates the concentration of iodide into the thyroid gland and plays an essential role in thyroid hormonogenesis. NIS is also expressed in several non-thyroid tissues, including lactating breast, stomach, intestine, salivary glands, kidney, ovary, testes, and choroid plexus [1–7]. The function of NIS in some of these sites, such as the reproductive tissues and the choroid plexus, is poorly defined. Human NIS exists as a homodimer comprised of a 643-residue multi-N-linked-glycosylated protein with thirteen putative alpha-helical transmembrane domains, with an extracellular N-terminus and intracellular C-terminus [8–11]. Rat NIS was first characterized in 1996 and reported to transport iodide with a stoichiometric ratio of 2 $Na^+$: 1 $I^-$ using a sodium gradient generated by the $Na^+/K^+$-ATPase [12, 13]. Human NIS is presumed to have the same stoichiometry as rat NIS, though this has never been experimentally confirmed. NIS exclusively transports monovalent anions, encompassing a wide range of substrates including $BF_4^-$, $TcO_4^-$, and $SCN^-$ (Table 1. References reporting uptake, inhibition, stoichiometry and $K_m$: [13–30]. References reporting fresh and seawater anion concentrations: [31–51].) [13–15]. Remarkably, the $Na^+$:anion stoichiometry with which these anions are transported is not always 2:1. Several substrates, such as $TcO_4^-$ and $ReO_4^-$, and the potent competitive substrate perchlorate ($ClO_4^-$), are reported to be transported with a 1:1 ratio [15, 52]. The exact mechanism of this stoichiometric shift is unknown. Mutagenesis and modeling studies indicate that glycine 93 (G93) and glutamic acid 94 (Q94) play a role in controlling substrate stoichiometry and ion coordination in conjunction with tryptophan 255 (W255) and tyrosine 259 (Y259) [52, 53]. Mutation of G93 to T, N, Q, E, or D shifts the stoichiometry of perchlorate transport from electroneutral to electrogenic [52]. However, our group and others hypothesize that G93X mutations disrupt Q94, which we model as part of the ion binding site [53].

Another unexplained phenomenon is the range of binding affinities NIS displays for various substrates. In rat NIS, iodide is transported with a $K_m$ value of 9.7–33 μM [13, 16, 17]. Other substrates are reported to have the following $K_m$ values in rat NIS: $ClO_4^-$: 0.59–3.8 μM, $ClO_3^-$: 486 μM, $SCN^-$: 38 μM, $SeCN^-$: 38 μM, $NO_3^-$: 770 μM, and $ReO_4^-$: 1.9 μM [13, 16, 18]. No mechanism has been identified to explain the different affinities for these chemically similar anions. Several studies have speculated that anion size plays a role, especially if selectivity and affinity are introduced in the ion binding pocket, where free movement of the ions may be restricted, and size may be an exclusion factor [13–15, 52–54].

In addition to its unique biology, NIS is of interest to the fields of oncolytic virotherapy, gene therapy, and cellular therapy where it has been used as a reporter gene to track the location of NIS-gene modified cells and the spread of therapeutic viruses or cells via nuclear imaging. The relatively promiscuous nature of NIS substrate selectivity means there are several high-affinity anionic substrates which can be used with different imaging modalities: $^{123}I^-$, $^{125}I^-$, $^{131}I^-$, $^{99m}TcO_4^-$, and $^{188}ReO_4^-$ for single photon emission computed tomography (SPECT); $^{124}I^-$ and $B^{18}F_4^-$ for positron emission tomography (PET), and $^{131}I^-$ for optical imaging via Cherenkov luminescence [55, 56]. NIS can also be used for targeted radiotherapy, as it

**Table 1. Chemical and kinetic properties and environmental abundance of NIS substrates and inhibitors.**

| Compound | Formula | MW (g/mol) | Molecular Volume (Å³) | Geometry | Stoichiometry Na⁺/anion | Reported $K_m$ in μM | Reported μM Concentration in Freshwater | Reported μM Concentration in Seawater |
|---|---|---|---|---|---|---|---|---|
| Bromide | $Br^-$ | 79.9 | 28.32 | spherical | ND | ND | 0.173–49.7 | 813.5–1001 |
| Iodide[1] | $I^-$ | 126.9 | 34.31 | spherical | 2:1 | 9.7–33 | 0.0016–0.015 | 0.14–0.46 |
| Astatide[1] | $At^-$ | 210.0 | 40.59 | spherical | ND | ND | ND | ND |
| Nitrate[1] | $NO_3^-$ | 62.0 | 41.24 | planar | 2:1 | 739–770 | 3.2 | 7.6 |
| Thiocyanate[1] | $SCN^-$ | 58.1 | 43.65 | linear | ≥2:1 | 20–96 | ND | 0.14–0.26 |
| Chlorate[1] | $ClO_3^-$ | 83.4 | 48.47 | trig. pyram. | ≥2:1 | 277–486 | 0.0012–0.53 | ND |
| Selenocyanate[1] | $SeCN^-$ | 104.9 | 50.18 | linear | 2:1 | 38 | ND | ND |
| Bromate | $BrO_3^-$ | 127.9 | 52.54 | trig. pyram. | ND | ND | 0.0008–10.8 | 0.0016 |
| Fluorosulfate[1] | $FSO_3^-$ | 99.1 | 54.27 | tetrahedral | ND | ND | ND | ND |
| *Perchlorate*[2] | $ClO_4^-$ | 99.4 | 56.55 | tetrahedral | 1:1 | 1.5–3.9 | 0.0005–0.6* | 0.0003* |
| Iodate | $IO_3^-$ | 174.9 | 58.54 | trig. pyram. | ND | ND | 0.00023–0.0019 | 0.114–0.343 |
| Metaperiodate | $IO_4^-$ | 213.9 | 66.61 | tetrahedral | ND | ND | ND | ND |
| *Hexafluorophosphate*[2] | $PF_6^-$ | 145.0 | 72.61 | octahedral | ND | ND | ND | ND |
| Pertechnetate[1] | $TcO_4^-$ | 162.0 | 72.89 | tetrahedral | 1:1 | 0.1 | ND | $3^{-6}$–0.009** |
| Perrhenate[1] | $ReO_4^-$ | 250.2 | 72.89 | tetrahedral | 1:1 | 1.9–2.3 | ND | 0.032–0.043*** |
| Tetrafluoroborate[1] | $BF_4^-$ | 86.8 | 73.2 | tetrahedral | ND | ND | ND | ND |
| Hexafluoroarsenate[2] | $AsF_6^-$ | 188.9 | 74.71 | octahedral | ND | ND | ND | ND |
| Hexafluoroantimonate[2] | $SbF_6^-$ | 235.8 | 86.98 | octahedral | ND | ND | ND | ND |

Compounds arranged by increasing molecular volume, calculated by Molinspiration Cheminformatics program Calculation of Molecular Properties and Bioactivity Score.

[1] indicates compound reported as a NIS substrate by either radiotracer uptake assays or electrophysiology.

[2] indicates compound reported to inhibit the ability of NIS to concentrate iodide, with italics indicating that the compound has been shown to be transported. All other compounds are transported by NIS to a varying degree. trig. pyram. = trigonal pyramidal; ND = no data

*These values may be due to industrial contamination of freshwater sources.

** As technetium.

*** As rhenium.

is able to concentrate the beta-emitting radioisotopes $^{131}I^-$, $^{186}ReO_4^-$, $^{188}ReO_4^-$, and alpha-emitting $^{211}At^-$ [57, 58].

Despite the growing utility of NIS as a nuclear imaging reporter, there remain several challenges to NIS imaging and therapy. Endogenous NIS expression, efflux of radiotracer from NIS-expressing cells, and sub-optimal NIS expression in target tissues decrease the efficacy of NIS imaging. The greatest problem lies with endogenous NIS expression. Concentration of radioisotopes in non-target tissues which naturally express NIS, such as the thyroid, salivary glands, and stomach, reduce the specificity and resolution of NIS imaging, and limit the safety of radiotherapy with NIS [59–64]. Several studies have explored ways to improve NIS imaging and treatment but to date, there has been limited progress in this area [65–68]. A NIS variant resistant to perchlorate inhibition would be particularly useful in this regard since it would remain active in the presence of Perchloracap®, an FDA approved perchlorate preparation which can eliminate background PET and SPECT signals due to radiotracer uptake by endogenous NIS [23, 66]. However, to date, there are no reports of NIS proteins with altered substrate specificity or inhibitor sensitivity which also maintain wild type transport kinetics.

Previous studies have reported differences in sodium and iodide binding and transport between mouse, rat, and human NIS. When expressed in COS-7 and HeLa cells, rat and mouse NIS were reported to concentrate 4-5x and 1.6x higher levels of iodide than human NIS, respectively [69, 70]. The majority of the observed effect could be explained by enhanced cell surface localization of the rodent proteins in the cell lines evaluated in those studies, which was 4-5x that of human NIS for the same amount of transfected DNA. An observed difference between mouse NIS and human NIS was that the $K_m$-$I^-$ is 2.5x lower in the mouse than human, whereas the values were very similar between rat and human NIS [69, 70]. Rat and mouse NIS share 87.3% and 86.2% amino acid sequence similarity with human NIS, respectively. Thus, even in these well-conserved proteins, there is evidence to suggest that some aspects of substrate translocation are different [69, 70].

Iodide concentration and availability differ across habitats and ecosystems, which raises the possibility that different species of animals may have NIS proteins with altered affinity for iodide, and by extension, other anionic NIS substrates [31]. We therefore assessed the NIS proteins from thirteen animal species for their ability to concentrate $^{125}I^-$ in the absence and presence of the NIS inhibitor perchlorate. Based on the disparate responses to perchlorate inhibition that were observed in minke whale (*Balaenoptera acutorostrata scammoni*) NIS, human (*Homo sapiens*) NIS, and zebrafish (*Danio rerio*) NIS, we subjected these NIS proteins to further assays assessing $^{125}I^-$ uptake in the presence of sixteen compounds previously reported to be NIS substrates and inhibitors. These assays revealed a wide range in substrate selectivity and inhibitor sensitivity across NIS proteins from different species. These findings indicate that evolutionarily-related NIS proteins have distinct transport properties and that small changes in the amino-acid sequence can have a profound effect on substrate selectivity and inhibitor sensitivity. Molecular modeling of NIS revealed highly conserved putative ion binding pockets, suggesting the residues responsible for the observed differences in transport behavior lie elsewhere in the protein. This opens an avenue for fine-tuning of NIS for radio-tracer uptake in the presence of competitive inhibitors such as perchlorate by modification of more peripheral protein residues which would not interfere with the anion binding to the putative binding sites. Further investigations are required to elucidate the residues and mechanism responsible for altered anion selectivity across the NIS proteins of different animal species.

## Materials and methods

### NIS protein selection

Experimentally-determined and predicted amino acid sequences of NIS proteins from different animal species were obtained from the NCBI Database and UniProt. Sequences were aligned to human NIS manually.

### Mammalian cell culture

HEK-293T (ATCC #ACS-4500) and HeLa (ATCC #CCL-2) human cell lines were obtained from ATCC (Manassas, VA) and were not authenticated but were tested for mycoplasma contamination. Cells used for all experiments were maintained in high-glucose Dulbecco's modified Eagle's medium (DMEM) (GE Healthcare, Chicago, IL, #SH30022.01) with 10% fetal bovine serum (FBS), 1% penicillin/streptomycin (P/S) and 1% Antibiotic-Antimycotic (A/A) (Gibco, Dun Laoghaire, Ireland, #15240062) and incubated at 37°C with 5% $CO_2$. Transduced HeLa cells were rinsed with phosphate buffered saline (PBS) and dissociated from the plastic substrate with Versene (Gibco #15040066) to avoid cleaving the extracellular loops of NIS.

## NIS construct generation

NIS plasmids containing *Homo sapiens*, *Papio anubis*, *Canis lupus familiaris*, *Sus scrofa*, *Rattus norvegicus*, and *Mus musculus* NIS cDNA were obtained from Imanis Life Sciences, LLC (Rochester, MN). pUC57 plasmids containing *Clupea harengus*, *Danio rerio*, *Xenopus laevis*, *Pelodiscus sinensis*, *Haliaeetus leucocephalus*, *Tursiops truncatus*, and *Balaenoptera acutorostrata scammoni* NIS cDNA flanked by MluI and NotI restriction sites were synthesized by GenScript (Piscataway, NJ). The NIS cDNAs from non-mammalian species were codon-optimized by GenScript for mammalian expression. All NIS sequences were cloned into a pHR-SFFV-HA-PGK-PURO transfer lentiviral vector derived from the pHR-SFFV-GFP-IRES-PGK-PURO transfer vector, a gift from Dr. Yasuhiro Ikeda. Each construct contains an HA tag immediately after the start codon flanked by BamHI and MluI restriction sites. Plasmids were Sanger sequenced to ensure sequence fidelity after cloning. The GFP-IRES sequence in the lentiviral vector was replaced with NIS. NIS does not have a leader sequence thus the HA tag could be added directly to the N-terminus. N-terminally tagged NIS constructs have been created before and expressed well [70].

## Transient transfection

Prior to the uptake assays, the amount of DNA necessary to achieve equal expression of each species of NIS protein was determined. For each construct, $1.0 \times 10^7$ HEK-293T cells were plated in a 15 cm dish. The next morning, the media was replaced with 15 ml DMEM + 1% heat-inactivated FBS. Next, 45.92 μg human NIS transfer vector, 36.74 μg minke whale NIS transfer vector, or 146.96 μg zebrafish NIS transfer vector were mixed with 500 μl OPTI-MEM (Gibco #31985070). For human-, minke whale-, or zebrafish NIS, 183.68, 146.96, or 293.92 μl of 1 mg/ml polyethylenimine (PEI) (1:4 DNA:PEI) (Polysciences, Warrington, PA, #23966–2) respectively was added drop-wise to the DNA/OPTI-MEM mix while vortexing to prevent DNA precipitation. After incubating at room temperature for 5 minutes, the DNA/PEI mix was added dropwise to the plate. Cells were incubated with DNA and PEI for 7 hours. Media was removed and replaced with pre-warmed 16 ml DMEM + 10% FBS + 1% P/S + 1% A/A.

## Lentiviral particle production and titration

Lentiviral production and transduction was approved by Mayo's Institutional Biosafety Committee. VSV-G pseudotyped lentiviral particles were generated in HEK-293T cells. The same transfection protocol described above was used with the following differences: $7.5 \times 10^6$ HEK-293 cells were plated in a 15 cm dish. 9.4 μg transfer vector, 9.4 μg GAG vector, and 3.2 μg VSV-G vector was mixed with 500 μl OPTI-MEM. 88 μl of 1 mg/ml PEI was added drop-wise to the DNA/OPTI-MEM mix while vortexing to prevent DNA precipitation. 7 hours after the addition of DNA/PEI, media was removed and replaced with pre-warmed 14 ml low glucose DMEM (Thermo Fisher, Waltham, MA, #10567–022) + 10% heat-inactivated FBS + 1% P/S + 25 mM HEPES (Gibco #15630080). 48 hours post infection, supernatant was filtered through a 0.45 μM filter, aliquoted, and frozen at -80ºC. Infectious titers were determined by flow cytometry (protocol can be found at: dx.doi.org/10.17504/protocols.io.bcbsisne) and the total lentiviral particle titers were determined by Lenti-X™ p24 Rapid Titer Kit (Clontech, Mountain View, CA, #632200).

## Stable cell line production

$2.5 \times 10^5$ HeLa cells were plated in six well plates. HeLa cells were infected at a multiplicity of infection (MOI) of 5 infectious viral particles per cell. After one week, cells were selected with

1.25 μg/ml puromycin for one week, by which time all non-transduced HeLa cells were dead. Puromycin-selected cells were maintained in 1.25 μg/ml puromycin. To equalize the percentage of live cells expressing HA-human-, HA-whale-, and HA-zebrafish NIS for comparative studies, cell lines were titrated with puromycin to determine a selection level which would equalize the percentage of cells expressing HA-tagged NIS such that protein expression levels could be normalized. HeLa-HA-human NIS and HeLa-HA-zebrafish NIS were treated with 5 μg/ml puromycin. The HeLa-HA-whale NIS cell line was treated with 3.25 μg/ml puromycin.

## Characterization of extracellular membrane-localized NIS protein expression by flow cytometry

To determine the relative expression of NIS proteins exclusively on the extracellular membrane of the cell, $1.5x10^5$ HeLa and HeLa-HA-NIS cells, in triplicate, were fixed in 500 μl 4% paraformaldehyde in PBS, then washed twice with 3 ml FACS buffer. Cells were stained with HA-tag (6E2) mouse mAb-Alexa Fluor 647 Conjugate (Cell Signaling Technology, Danvers, MA, #3444) 1:40 dilution for 1 hour at room temperature in 100 μl FACS buffer. Cells were washed twice with 3 ml cold FACS buffer. Singlets were isolated and gated on non-transduced HeLa cells incubated with antibody.

## Radioactive substrate uptake assays

$8x10^5$ HeLa-HA-NIS cells were plated in six well plates (Falcon #353046) 16 hours prior to $^{125}I$, $^{99m}TcO_4$, and $B^{18}F_4$ uptake assays. Three independent replicates of six wells were used per condition. Immediately prior to the uptake assay, cells were incubated at 37˚C in 1 ml uptake buffer (Hank's balanced salt solution (HBSS) + 10 mM HEPES, pH 7.4) with or without 50 μM $NaClO_4^-$. Radioactive substrate solutions were prepared immediately prior to each assay. $Na^{125}I$ in 0.1 M NaOH (Perkin Elmer, Waltham, MA), $Na^{99m}TcO_4$ (Mayo Clinic's Nuclear Medicine Pharmacy), or $NaB^{18}F_4$ (Dr. Timothy DeGrado, Mayo Clinic) was diluted in uptake buffer such that 5 μl contained ~600,000, ~750,000, or ~1,000,000 counts per minute via gamma counter for $^{125}I^-$, $^{99m}TcO_4^-$, or $B^{18}F_4^-$ respectively. 5 μl of isotope was carefully added to each well. Cells were incubated at 37˚C for 50 minutes. Wells were washed twice with 2 ml ice cold uptake buffer. Cells were incubated at 37˚C for 10 minutes with pre-warmed 1 ml 1M NaOH to lyse. Cells were shaken for 5 minutes to further lyse. Lysate was collected and transferred to tubes (Sarstedt, Nümbrecht, Germany, #55.476.005 & 65.809) for gamma counting via a Wallac 1480 gamma counter with a sodium iodide detector. Counts for $^{99m}TcO_4^-$ and $B^{18}F_4^-$ were corrected for radioactive decay due to the short half-lives of the isotopes.

Uptake assays with transfected HEK293T cells were performed as above with the following differences: Prior to cell plating, wells were coated with 1 mg/ml fibronectin (Sigma #F1141-1MG) and allowed to dry for 45 minutes. In the evening, $4.5x10^5$ transfected cells were plated in 12 well plates (Corning Costar #3527). 16 hours later, wells were loaded with ~250,000 counts per minute $^{125}I^-$. Three independent replicates of two wells were used per condition.

## Competitive substrate uptake assays

Assays were performed as described in 'Radioactive substrate uptake assays' with the following differences: $1.5x10^5$ HeLa-HA-NIS cells were plated in six well plates three days prior to the competitive substrate uptake assays. Two independent replicates of three wells were used for each condition. Immediately prior to the uptake assay, cells were incubated at 37˚C in 1 ml uptake buffer with or without added cold substrate or inhibitor compounds. Separately, $Na^{125}I$ in 0.1 M NaOH was diluted in uptake buffer such that 50 μl contained ~600,000 counts per

minute via gamma counter. 50 μl of Na$^{125}$I was carefully added to each well. Gamma counting performed with GMI ISO Data 20/10 Multiwell gamma counter. Potassium salts of BrO$_3$, ClO$_3$, FSO$_3$, IO$_3$, NO$_3$, PF$_6$, and SeCN, and sodium salts of AsF$_6$, BF$_4$, Br, ClO$_4$, I, IO$_4$ ReO$_4$, SbF$_6$, and SCN were obtained from Sigma-Aldrich (St. Louis, MO).

### Iodide uptake kinetic assay

Iodide uptake assays to evaluate K$_m$ and V$_{max}$ were performed following the protocol described by [71] with the following modifications: 2x10$^4$ HeLa-HA-NIS cells were plated in 96 well plates (Corning Falcon #351172) three days prior to the iodide uptake assay. Cells were washed with 200 μl 37˚C uptake buffer. Cells were then incubated with 0, 0.39, 0.78. 1.56, 3.125, 6.25, 12.5, 25, 50, and 100 μM KI in 100 μl 37˚C uptake buffer for 4 minutes. Cells were washed twice with 200 μl ice-cold uptake buffer then dried by inverting the plates and tapping them against absorbent paper. To assess the amount of iodide concentrated by the above assay, the colorimetric iodide assay was performed immediately afterwards and was conducted exactly as described by [72]. Three independent replicates were performed with 8 wells per condition. Ammonium cerium(IV) sulfate hydrate, concentrated sulfuric acid (99% pure), arsenic(III) oxide, and sodium hydroxide were obtained from Sigma-Aldrich.

### Software and statistics

Flow cytometry data were analyzed using FloJo 10. Figures and statistics were generated using GraphPad Prism 8.1.1 and Photoshop CC. p-values were calculated using an unpaired two-tailed $t$-test with Welch's correction. IC$_{50}$ values were calculated via non-linear regression using the equation setting: {inhibitor} vs. normalized response—variable slope—inhibition. K$_m$ and V$_{max}$ values were calculated by fitting the data via the least squares (ordinary) fit method for non-linear regression with the Michaelis-Menten equation. Molecular volume values were generated using the Molinspiration Cheminformatics program *Calculation of Molecular Properties and Bioactivity Score* (http://www.molinspiration.com/). Evolutionary distance was estimated using Time Tree, developed by the Institute for Genomics and Evolutionary Medicine Center of Biodiversity at Temple University (http://www.timetree.org/) [73]. Protein similarity and identity were assessed by EMBOSS Needle Pairwise Sequence Alignment (Protein) with default settings from the European Bioinformatics Institute (https://www.ebi.ac.uk/Tools/psa/emboss_needle). Software using for molecular modeling is described below.

### Creation of NIS models

Detailed methods for the creation of the human NIS model are available online (dx.doi.org/10.17504/protocols.io.bb46iqze). Briefly, a semi-occluded homology model of hNIS (transmembrane domains 1–12) was prepared based on two templates: vSGLT, a sodium galactose transporter from *Vibrio haemolyticus* (PDB code 3DH4 [74]) and Mhp1, a Na$^+$ coupled hydantoin transporter from *Microbacterium liquefaciens* (PDB code 4D1B [75]). Both templates feature the common LeuT-fold architecture identified for the first time in the crystal structure of the bacterial leucine transporter LeuT [76]. NIS is expected to have the same architecture as a homologous protein to the human sodium glucose transporters (hSGLT family), which are evolutionary related to vSGLT [24, 77]. Sequence alignments were performed with Modeller 9.18 with additional manual alterations [78]. 500 3D models were generated with the automodel function in Modeller 9.18, with alpha helical constraints imposed on human NIS residues 85–99, 279–296, 383–401, 409–425, and 428–438 for better reproduction of helical transmembrane domains. The quality of these models was evaluated with pdfpdb [78], DOPE [79], and GA341 [80] scores. 5–10 structures with the lowest DOPE and highest GA341 scores were

overlapped with the templates to assess similarity and structural integrity. Three structures with the best side chain overlap with known residues of interest in vSGLT and Mhp1 were selected for side chain relaxation with ROSETTA MP [81]. 2000 decoy structures were generated for each of the 3 structures selected and were scored with the mpframework_s-mooth_fa_2012 function [81]. These were then clustered with the cluster tool in ROSETTA. The structure at the center of the largest lowest energy cluster was chosen as a representative human NIS model.

To generate the minke whale NIS and zebrafish NIS models, the human NIS model developed above was used as a template. The sequence alignment was performed manually. The sequences were threaded onto the human NIS model via the SWISS-MODEL server [82]. These structures were then subjected to side chain relaxation and clustering in ROSETTA MP as described above. The centers of the lowest energy clusters were selected for comparison with hNIS for the purposes of this work. The key residues involved in ion binding in NIS were identified from Molecular Dynamics simulations.

## Results

### Selection of NIS protein panel

NIS amino acid sequences from fifty-three animal species were aligned and examined (alignment available upon request). Thirteen species were selected due to their position in evolutionary history, interesting attributes, or use as a model species (Table 2). Atlantic herring (*C. harengus*), minke whale (*B. acutorostrata scammoni*), and bottle-nosed dolphin (*T. truncatus*) NIS were selected to explore marine species, as seawater has the highest concentration of iodide [31, 34]. Bottle-nosed dolphin NIS was also selected to examine the functional impact of the reported 48 amino acid deletion after the putative site of transmembrane domain 11. Zebrafish (*D. rerio*) NIS was selected due to its use as a model species and its freshwater habitat. African clawed frog (*X. laevis*), Chinese softshell turtle (*P. sinensis*), and bald eagle (*H. leucocephalus*) NIS were selected as evolutionary representatives of amphibians, reptiles, and

**Table 2. Properties of NIS proteins investigated.**

| Latin Name | Species | Length | % Similarity | % Identity | NCBI Reference Sequence # |
|---|---|---|---|---|---|
| *H. sapiens* | Human | 643 | - | - | NP_000444.1 |
| *P. anubis* | olive baboon | 643 | 98.4 | 96.9 | XP_003915201.1* |
| *B. acutorostrata scammoni* | minke whale | 642 | 91.1 | 85.8 | XP_007195539.1 |
| *C. lupus familiaris* | Dog | 642 | 90.5 | 85.1 | XP_541946.3 |
| *R. norvegicus* | Rat | 618 | 87.3 | 81.4 | NP_443215.1* |
| *M. musculus* | Mouse | 618 | 86.2 | 77.6 | NP_444478.2* |
| *T. truncatus* | bottle-nosed dolphin | 597 | 84.8 | 79.3 | XP_004323920§ |
| *S. scrofa* | Pig | 665 | 83.6 | 83.8 | NP_999575.1* |
| *X. laevis* | African clawed frog | 645 | 75.5 | 69.0 | NP_001086891.1 |
| *H. leucocephalus* | bald eagle | 652 | 74.7 | 64.8 | XP_010561595.1 |
| *D. rerio* | zebrafish | 601 | 71.9 | 57.6 | NP_001082860.1 |
| *C. harengus* | Atlantic herring | 580 | 69.9 | 54.9 | XP_012694952.1 |
| *P. sinensis* | Chinese softshell turtle | 515 | 53.4 | 46.0 | XP_014433638.1 |

Species are ordered by '% Similarity' as evaluated at the amino acid level by EMBOSS Needle Pairwise Sequence Alignment (Protein).

*Proteins used in this study have the following differences: olive baboon: Y81H, I432V, G507A, E600K, ETNL640-643RQTS; rat and mouse: A614V; pig: S447P, K588E, 620–665.

§Sequence no longer available.

birds. Mouse (*M. musculus*), rat (*R. norvegicus*), pig (*S. scrofa*), dog (*C. lupus familiaris*), olive baboon (*P. anubis*), and human (*H. sapiens*) NIS were included due to their use as model organisms and as representative mammals.

NIS amino acid sequences were highly conserved, especially in putative transmembrane domains (Fig 1, underlined residues in human NIS. Fig 1 shows the amino acid alignment between human NIS, minke whale NIS, and zebrafish NIS, as these three became the focus of this study. The alignment of all thirteen protein sequences is provided in S1 Fig). The majority of the residue differences were found in the N-terminus, extracellular loop 1, and the C-terminus, which contains a currently structurally undefined thirteenth transmembrane domain and the putatively unstructured C-terminal tail.

## Expression of extracellular membrane-localized NIS proteins

All cell lines transduced with HA-tagged-NIS-expressing lentiviral vectors survived 1.25 µg/ml puromycin selection, but not all NIS proteins were well-tolerated by the HeLa cells. Atlantic herring NIS and Chinese softshell turtle NIS appeared to be toxic to cells, as a low percentage of cells expressed Atlantic herring NIS and Chinese softshell turtle NIS (Fig 2A), and the level of expression was low (Fig 2B). This toxicity may arise from a misfolded protein response due to the non-mammalian nature of these proteins, but this was not tested. The remaining cell lines expressed NIS on the majority of transduced cells, though expression was negligible or low for bald eagle NIS and bottle-nosed dolphin NIS. Low bottle-nosed dolphin NIS expression is likely due to the missing 48 residues. The sequence was later withdrawn from the NCBI database, suggesting the missing region was an error in sequencing, not a biological difference in bottle-nosed dolphin NIS.

## Characterization of iodide uptake and perchlorate inhibition in NIS proteins from various animal species

All NIS expressing cell lines were subjected to $^{125}$I⁻ uptake assays in the absence or presence of 100 µM perchlorate. Values shown are not normalized for protein expression level.

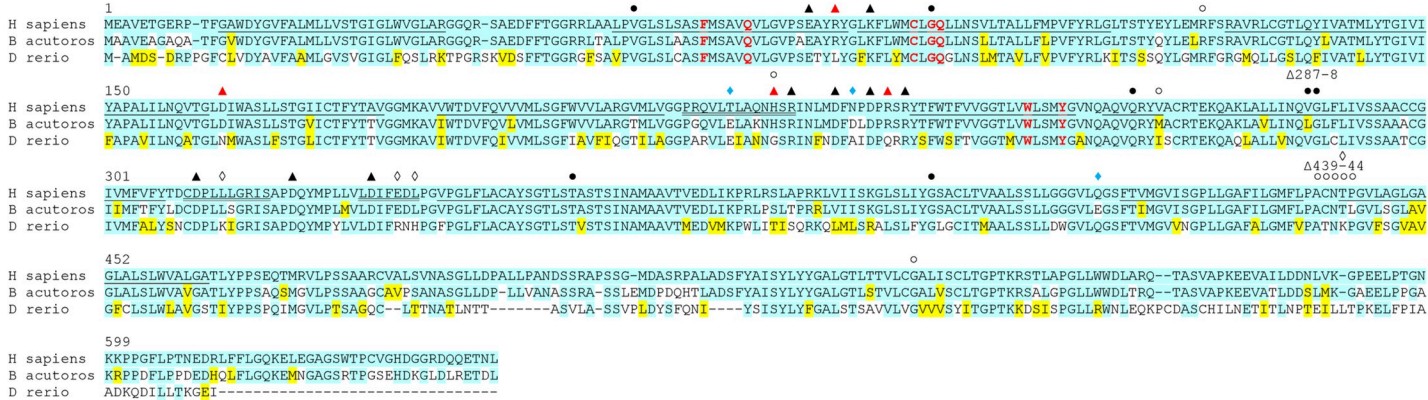

**Fig 1. Amino acid sequence alignment between the NIS proteins of *H. sapiens* (human), *B. acutorostrata scammoni* (minke whale), and *D. rerio* (zebrafish).** Cyan highlighting indicates absolute conservation to human NIS. Yellow indicates similar residue to human NIS. Underline indicates putative transmembrane domain in human NIS, only TM1-12 are indicated. Closed circles indicate site of a mutation known to cause a transport defect in humans [22]. Open circles indicate site of a mutation known to cause membrane trafficking defect in humans [22]. Black triangles indicate a charged residue where mutation to alanine significantly reduces iodide uptake in human NIS [71]. Red triangles indicate a charged residue where mutation to alanine significantly reduces iodide uptake in human NIS and this residue is not charged in zebrafish NIS [71]. Open diamonds indicate additional positively charged residues in zebrafish NIS. Blue diamonds indicate additional negatively charged residues in minke whale NIS. Bold red lettering indicates residue reported to be involved in stoichiometry control and translocation dynamics [52–53]. Numbering follows human NIS.

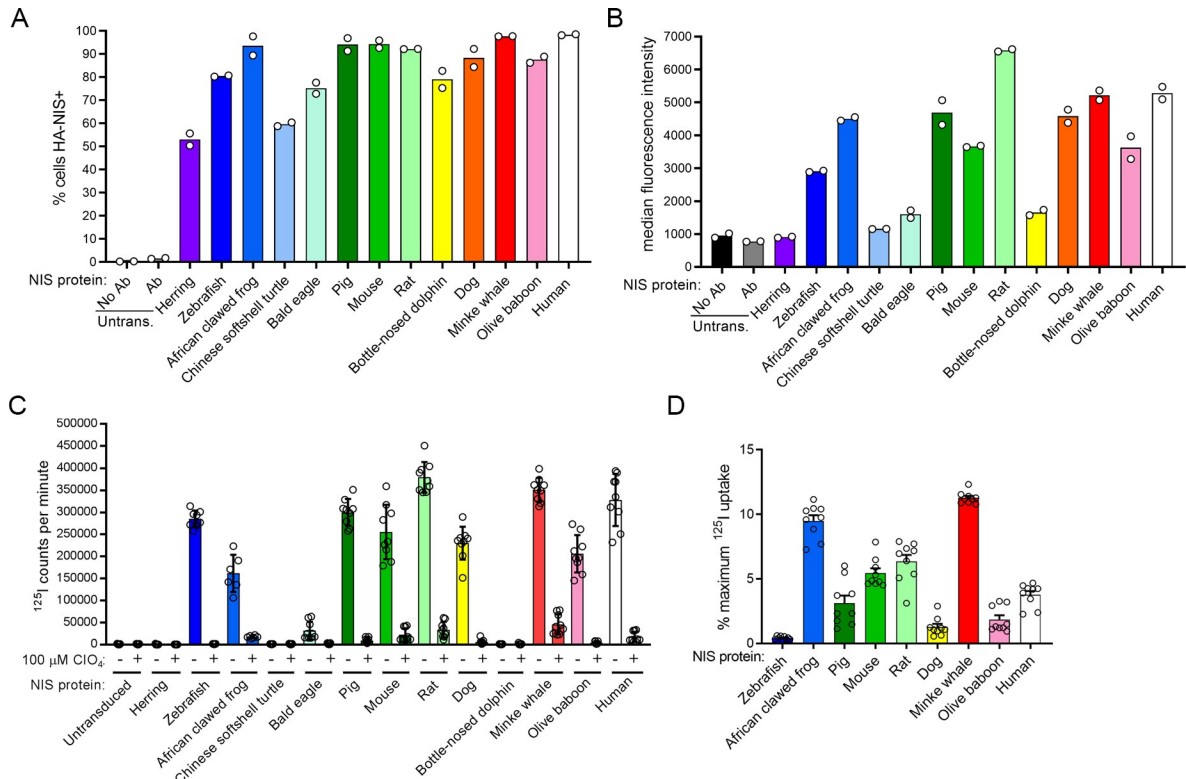

**Fig 2. Characterization of HA-NIS protein expression and iodide transport.** Species are ordered in ascending evolutionary proximity to humans as determined by TimeTree (pig, mouse, rat, and dog diverged equidistantly) [73]. (A) Percentage of 1.25 μg/ml puromycin-selected cells which were positively-stained with α-HA-AlexaFluor647 antibody. (B) Median fluorescence intensity of 1.25 μg/ml puromycin-selected cells which were stained with α-HA-AlexaFluor647 antibody. Values are averages of duplicate assays. (C) Absolute uptake value of each NIS protein in the absence or presence of 100 μM perchlorate ($ClO_4$), which has not been equalized to protein expression. Values are averages of triplicate assays with standard deviation. (D) Percentage of maximum $^{125}I^-$ uptake maintained in the presence of 100 μM perchlorate. Untrans. = untransduced, Ab = antibody.

Untransduced HeLa cells did not concentrate iodide. Atlantic herring NIS, Chinese soft-shelled turtle NIS, bald eagle NIS, and bottle-nosed dolphin NIS either did not concentrate iodide or concentrated iodide to a negligible degree (Fig 2C). The remaining NIS proteins concentrated iodide to appreciable levels and all were significantly inhibited by perchlorate, though the proteins differed substantially in their relative sensitivity to perchlorate inhibition. Minke whale NIS (wNIS) was the most resistant to perchlorate inhibition, as it maintained 13.1% of its maximum $^{125}I^-$ uptake activity in the presence of perchlorate. Zebrafish NIS (zNIS) was the most sensitive to perchlorate inhibition, as it maintained only 0.44% of its maximum $^{125}I^-$ uptake activity under the same conditions (Fig 2D).

## Characterization of extracellular membrane-localized human-, minke whale-, and zebrafish NIS after transient transfection or prolonged puromycin selection of lentivirally transduced cells

The previous experiments revealed wNIS and zNIS to be the least and most sensitive to perchlorate inhibition, respectively. Therefore, we focused on these two species of NIS protein, with human NIS (hNIS) as an intermediate inhibition representative. Transient transfections of HEK293T cells produced cell lines with ~55% of cells expressing HA-tagged NIS 48 hours post-transfection (Fig 3A). Expression levels were roughly equal, with HA hNIS expressing

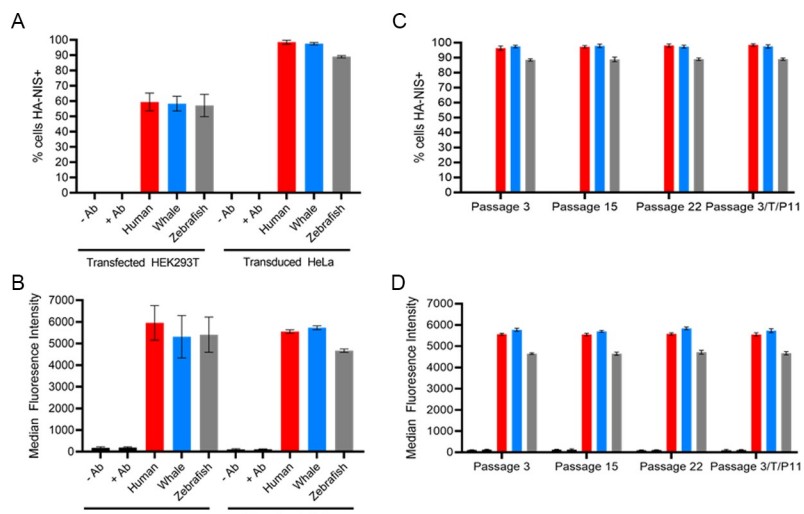

**Fig 3. Flow cytometry characterization of transfected HEK293T cells and transduced HeLa cells expressing HA-human NIS, HA-minke whale NIS, and HA-zebrafish NIS.** (A) Percentage of cells positively stained with α-HA-AlexaFluor647 antibody; 24-hour post transfection HEK293T cells compared to puromycin selected lentivirally-transduced HeLa cells. (B) Mean fluorescence intensity of HEK293T transfected cells or transduced HeLa cells stained with α-HA-AlexaFluor647 antibody. (C) Percentage of puromycin selected lentivirally transduced HeLa cells over several passages which were stained with α-HA-AlexaFluor647 antibody. 3/T/P11 indicates cells which were frozen at passage 3 post-puromycin selection, thawed, and grown out to passage 11. (D) Median fluorescence intensity of puromycin selected lentivirally transduced HeLa cells over several passages which were stained with α-HA-AlexaFluor647 antibody. Values are averages of triplicate assays with standard deviation. -/+ Ab refers to incubation of antibody with nontransfected/untransduced parental cell line.

slightly better, though expression levels were variable experiment-to-experiment (Fig 3B). To create cell lines with more stable expression, the lentivirally transduced HeLa cells initially selected with 1.25 µg/ml puromycin were subjected to a puromycin titration to identify the concentration of puromycin required to achieve near-equal expression between the three NIS constructs. The resulting cell lines expressed HA-hNIS in 95.2% of cells, HA-wNIS in 97.2%, and HA-zNIS in 89.4% (Fig 3A). Though the fluorescence intensity differed between the three cell lines, with wNIS expressed to the greatest degree, these values allowed us to subsequently compare the absolute ability of each protein to transport a given radioisotope substrate via normalization to hNIS (Fig 3B). The percentage of cells expressing HA-NIS and the levels of expression remained stable over many passages (Fig 3C and 3D).

## Comparison of $^{125}I^-$, $^{99m}TcO_4^-$, and $B^{18}F_4^-$ uptake and perchlorate inhibition in human-, minke whale-, and zebrafish NIS

Radioactive substrate uptake assays with transiently transfected HEK293T cells and HeLa-HA-hNIS, -HA-wNIS, and -HA-zNIS revealed all three proteins concentrate $^{125}I^-$ to similar levels (Fig 4A and 4B), with each protein concentrating equal levels of iodide after expression normalization (Fig 4E and 4F). All three proteins were sensitive to inhibition with 50 µM perchlorate (Fig 4E and 4F). zNIS was the most inhibited protein, with only 0.41% of maximum $^{125}I^-$ uptake activity maintained in the presence of perchlorate. hNIS retained 4.02% of maximum $^{125}I^-$ uptake activity, whereas wNIS retained the most, with 12.50% (Fig 4J). This inhibition trend was repeated with pertechnetate ($^{99m}TcO_4^-$) and tetrafluoroborate ($B^{18}F_4^-$) radioisotope uptake assays. All three proteins concentrated pertechnetate (Fig 4C), though wNIS transported 14.32% less than hNIS and zNIS transported 38.73% less than hNIS after expression normalization (Fig 4G). With 50 µM perchlorate treatment, hNIS maintained 2.92%

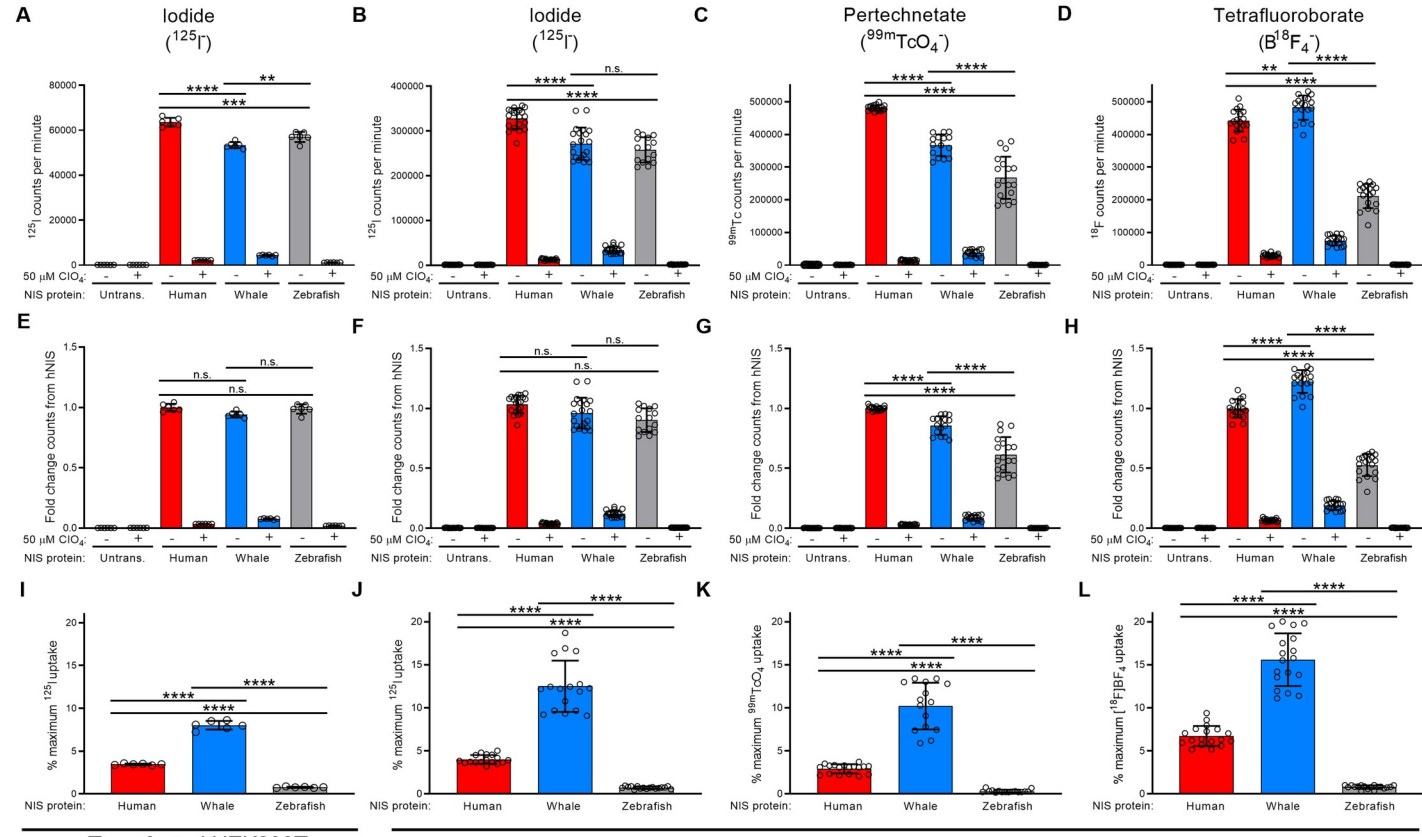

**Fig 4. Characterization of radioisotope substrate uptake in HeLa cells lentivirally transduced with HA-human NIS, HA-minke whale NIS, or HA-zebrafish NIS.** Raw uptake value of (A-B) $^{125}I^-$, (C) $^{99m}TcO_4^-$, or (D) $B^{18}F_4^-$ in the absence or presence of 50 μM perchlorate ($ClO_4$) in transfected or transduced cells. (E-H) Uptake of each radioisotope normalized to hNIS uptake via cell surface protein expression. (I-L) The percentage of maximum $^{125}I^-$ uptake activity (uptake at 0 μM $ClO_4$) maintained in the presence of 50 μM perchlorate ($ClO_4$). Values are averages of triplicate assays with standard deviation. n.s. indicates not significant; * indicates p < 0.05; ** indicates p < 0.01; *** indicates p < 0.001; **** indicates p < 0.0001; Untrans. = untransfected or untransduced.

maximum $^{99m}TcO_4^-$ uptake activity, wNIS maintained 10.20%, and zNIS maintained 0.32% (Fig 4K). wNIS transported the 22.45% more tetrafluoroborate than hNIS while zNIS transported 47.28% less than hNIS after expression normalization (Fig 4H). In the presence of 50 μM perchlorate, hNIS maintained 6.73% maximum $B^{18}F_4^-$ uptake activity, wNIS maintained 15.59%, and zNIS maintained 0.81% (Fig 4L).

## Perchlorate IC$_{50}$ determination for human-, minke whale-, and zebrafish NIS

After observing the differential response of each NIS protein to perchlorate inhibition, a perchlorate titration was performed to determine the IC$_{50}$ value. At the lowest concentration tested, 6 nM, zNIS lost 11.61% of maximum $^{125}I^-$ uptake activity. The IC$_{50}$ values for perchlorate were determined as follows: hNIS: 1.566 μM, wNIS: 4.599 μM, zNIS: 0.081 μM (Fig 5). The value obtained for hNIS is similar to other reported values of 0.488 μM and 1.27 μM [83, 84].

## Comparison of substrate selectivity and inhibitor sensitivity by human-, minke whale-, and zebrafish NIS

To further investigate the differences observed between the three NIS proteins in the presence of perchlorate, we subsequently tested fifteen additional monovalent anions for their ability to

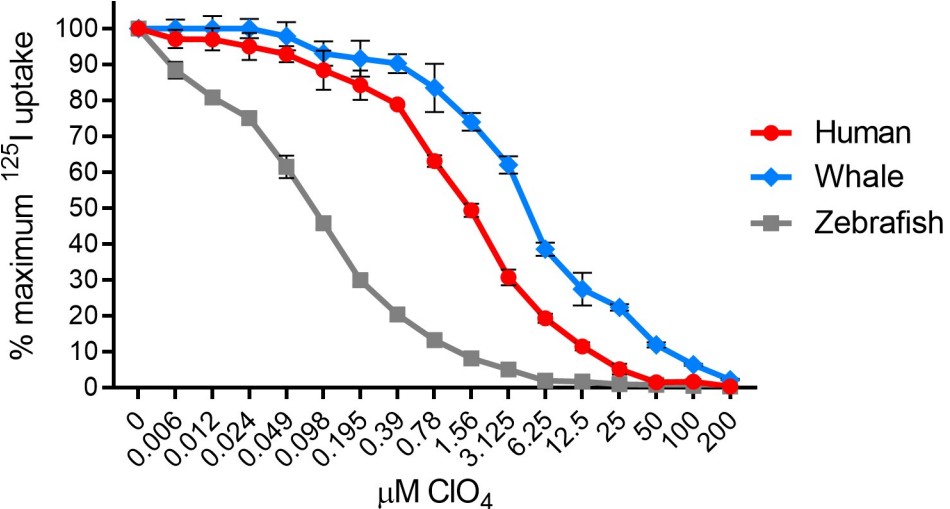

**Fig 5. Perchlorate IC$_{50}$ determination for HA-human-, HA- minke whale-, and HA-zebrafish NIS.** Lentivirally transduced HeLa cells were incubated with $^{125}$I$^-$ with increasing concentration of perchlorate (ClO$_4^-$) for 50 minutes prior to two washes with cold buffer. IC$_{50}$ values were determined via non-linear regression using the equation setting: {inhibitor} vs. normalized response—variable slope–inhibition in Prism 8. Human NIS ClO$_4^-$ IC$_{50}$ = 1.566 μM. Minke whale NIS ClO$_4^-$ IC$_{50}$ = 4.566 μM. Zebrafish NIS ClO$_4^-$ IC$_{50}$ = 0.081 μM. Values are average of triplicate assays with standard deviation.

interfere with NIS-mediated $^{125}$I$^-$ uptake. The sixteen anions tested included: twelve known NIS substrates (BF$_4^-$, Br$^-$, BrO$_3^-$, ClO$_3^-$, FSO$_4^-$, I$^-$, IO$_3^-$, IO$_4^-$, NO$_3^-$, ReO$_4^-$, SeCN$^-$, SCN$^-$), two potent competitive substrates (ClO$_4^-$ and PF$_6^-$), of which ClO$_4^-$ is considered the standard NIS inhibitor, one strong inhibitor (AsF$_6^-$), and one weak inhibitor (SbF$_6^-$) at concentrations ranging from 0.39–200 μM (Fig 6). Parallel data for NIS proteins from six additional animal species shown in Fig 2D are available in S2 Fig. NIS-negative HeLa cells did not concentrate iodide under any of the conditions tested.

Concentration of $^{125}$I$^-$ by hNIS was not inhibited or was minimally inhibited by Br$^-$, NO$_3^-$, BrO$_3^-$, IO$_4^-$, and SbF$_6^-$ (Fig 6A). hNIS was more sensitive to IO$_3^-$ and ClO$_3^-$. Cold iodide was competitive with loaded $^{125}$I$^-$ at 6.25 μM, with 200 μM I$^-$ reducing uptake by 75.3%. SCN$^-$ and SeCN$^-$ behaved much like I$^-$ despite their linear molecular geometry. The previously described NIS substrates FSO$_4^-$, ReO$_4^-$, and BF$_4^-$, reduced $^{125}$I$^-$ uptake more than any other substrate, almost eliminating $^{125}$I$^-$ uptake at 200 μM (92.9%, 98.1% and 96.9% inhibition, respectively). The previously described NIS inhibitors ClO$_4^-$, PF$_6^-$, and AsF$_6^-$ behaved as expected, with 0.39 μM compound reducing $^{125}$I$^-$ uptake by 39.6%, 29.9%, and 15.1%, respectively, and 200 μM reduced uptake to less than 1.25% of the maximum $^{125}$I$^-$ uptake activity for all three compounds.

Concentration of $^{125}$I$^-$ by wNIS was not inhibited or was minimally inhibited by BrO$_3^-$, NO$_3^-$, Br$^-$, IO$_4^-$, and SbF$_6^-$ (Fig 6B). 200 μM IO$_3^-$ inhibited wNIS by 15.6%. Cold iodide was competitive with loaded $^{125}$I$^-$ at 6.25 μM, with 200 μM I$^-$ reducing uptake by 60.8%. ClO$_3^-$, SCN$^-$, SeCN$^-$, and FSO$_4^-$ behaved similarly, reaching 70–80% inhibition at 200 μM. BF$_4^-$ did not inhibit uptake as well as ReO$_4^-$ at low concentrations, but both reduced $^{125}$I$^-$ to <10% at 200 μM. wNIS was more resistant than hNIS to inhibition by ClO$_4^-$, PF$_6^-$, and AsF$_6^-$ at low concentrations, but uptake was nearly abolished at 200 μM (<2.8% maximum $^{125}$I$^-$ uptake activity).

Compared to hNIS and wNIS, zNIS was more sensitive to competitive inhibition by every compound tested (Fig 6C). zNIS was modestly inhibited by Br$^-$, BrO$_3^-$, and NO$_3^-$. Unlike hNIS

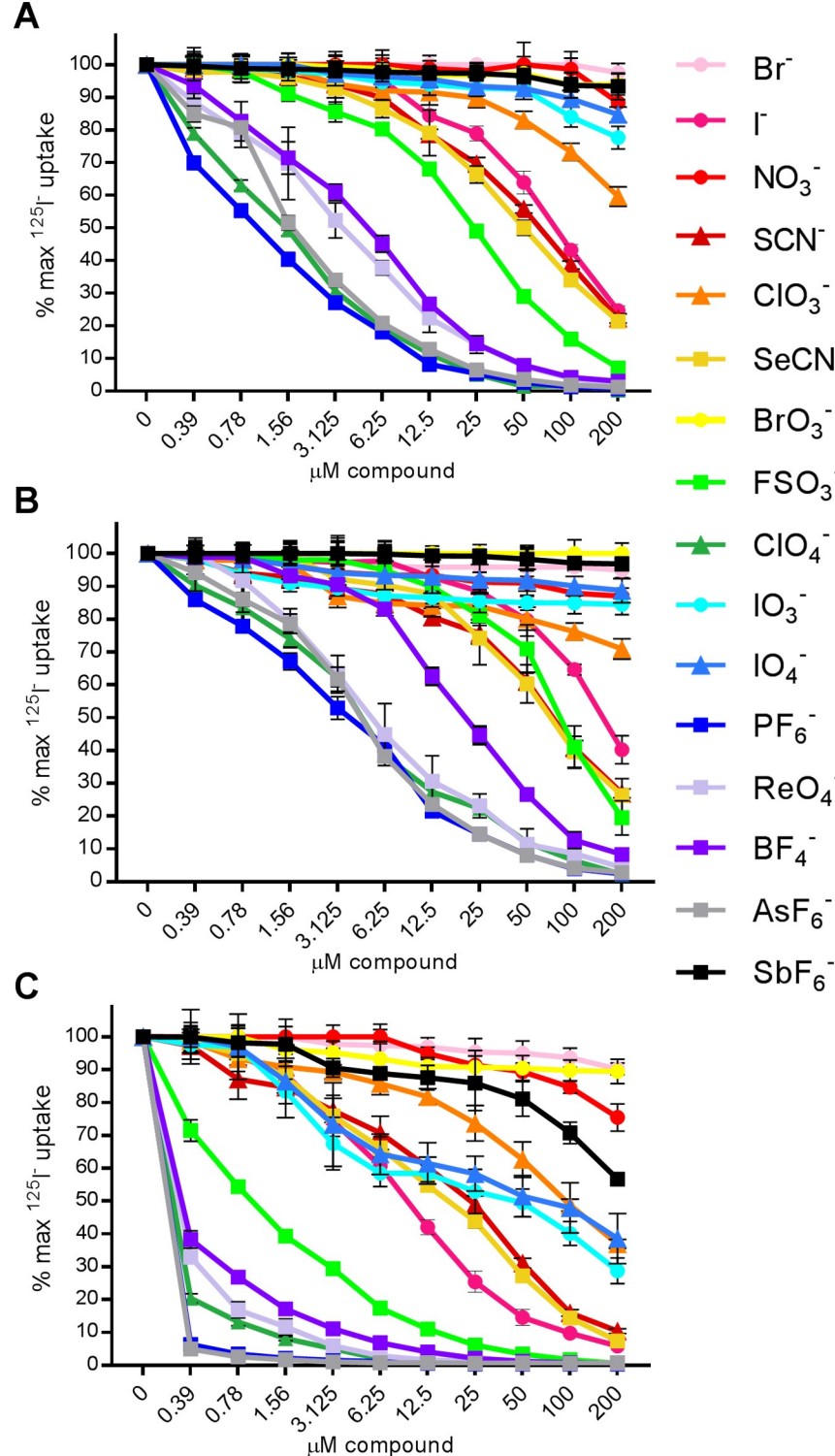

**Fig 6. Competitive substrate inhibition assays.** $^{125}I^-$ uptake with competitive substrates in HeLa cells transduced with (A) HA-human NIS, (B) HA-whale NIS, or (C) HA-zebrafish NIS. Data shown as the percentage of maximum $^{125}I^-$ uptake activity (0 μM compound) maintained in the presence of increasing concentrations of substrate or inhibitor. Circular markers indicate naturally occurring anions. Triangular markers indicate anions which may occur naturally at low levels or are generated inside the organism. Square markers indicate anions not found naturally. Values are averages of duplicate assays with standard deviation.

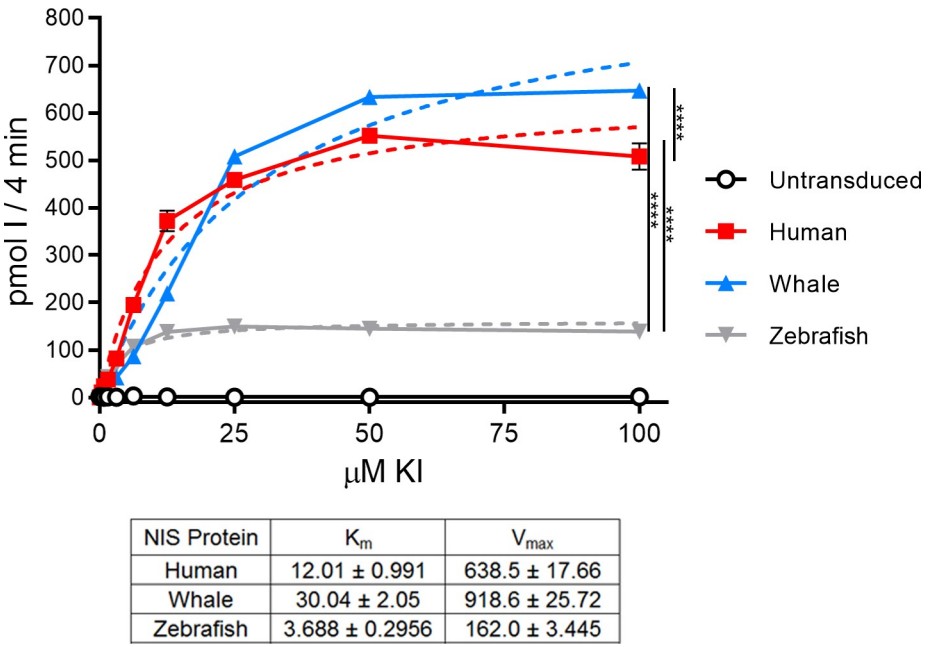

| NIS Protein | $K_m$ | $V_{max}$ |
|---|---|---|
| Human | 12.01 ± 0.991 | 638.5 ± 17.66 |
| Whale | 30.04 ± 2.05 | 918.6 ± 25.72 |
| Zebrafish | 3.688 ± 0.2956 | 162.0 ± 3.445 |

**Fig 7. Kinetic analysis of HA-human NIS, HA-minke whale NIS, and HA-zebrafish NIS.** Picomoles of iodide concentrated into HeLa cells lentivirally transduced with HA-NIS proteins after 4 minutes at 37˚C. Uptake values were normalized to cell surface protein expression. Solid lines and symbols indicate experimentally derived uptake values. Dashed lines indicate predicted Michaelis-Menten equation curves determined with the least squares (ordinary) fit method in Prism. $K_m$ and $V_{max}$ values are represented in the table. Units for $K_m$ are µM; units for $V_{max}$ are pmol $I^-$/4 minutes. Values are averages of triplicate assays with standard error. p-values calculated with experimental values at 50 µM KI; **** indicates $p < 0.0001$.

and wNIS, zNIS was sensitive to significant competitive inhibition by $SbF_6^-$, $IO_3^-$, $IO_4^-$, and $ClO_3^-$. Cold iodide was competitive with loaded $^{125}I^-$ after 0.78 µM, with 200 µM $I^-$ reducing uptake by 94.2%. $SCN^-$ and $SeCN^-$ behaved much like $I^-$. $FSO_4^-$ was the next most potent, behaving much like $ClO_4^-$ in hNIS. $ReO_4^-$ and $BF_4^-$ significantly reduced $^{125}I^-$ uptake, eliminating it at 50 µM and 100 µM, respectively. zNIS was much more sensitive to $ClO_4^-$, $PF_6^-$, and $AsF_6^-$ inhibition than hNIS or wNIS, with 0.39 µM compound reducing $^{125}I^-$ uptake over 90%.

## Iodide transport kinetics analysis of human-, minke whale-, and zebrafish NIS

To identify a potential explanation for the differences observed in the competitive uptake assays, we performed kinetic analysis of HA-tagged human-, whale-, and zebrafish NIS iodide uptake assays over a range of KI concentrations (Fig 7). Uptake values were normalized to hNIS via cell surface protein expression. The table in Fig 6 indicates the $K_m$ and $V_{max}$ values of each protein as calculated using standard Michaelis-Menten kinetics. wNIS had the lowest iodide binding affinity ($K_m$-I = 30.04 ± 2.05 µM), but the highest maximal transport velocity ($V_{max}$-I = 918.6 ± 25.72 pmol $I^-$ 4min$^{-1}$). hNIS continued to be the intermediate representative with $K_m$-I = 12.01 ± 0.991 µM and $V_{max}$-I = 638.5 ± 17.66 pmol $I^-$ 4min$^{-1}$. zNIS had the highest $K_m$ ($K_m$-I = 3.688 ± 0.2956 µM), but the lowest maximal transport velocity ($V_{max}$-I = 162.0 ± 3.445 pmol $I^-$ 4min$^{-1}$). hNIS binds iodide 2.5x more efficiently than wNIS, but transports it at a rate 1.44x lower than wNIS. zNIS binds to iodide with the greatest affinity of the proteins examined in this study, 8.1x greater than the wNIS $K_m$ and 3.26x greater than the hNIS $K_m$, but transports iodide at a lower rate than the other two mammalian proteins: 5.67x

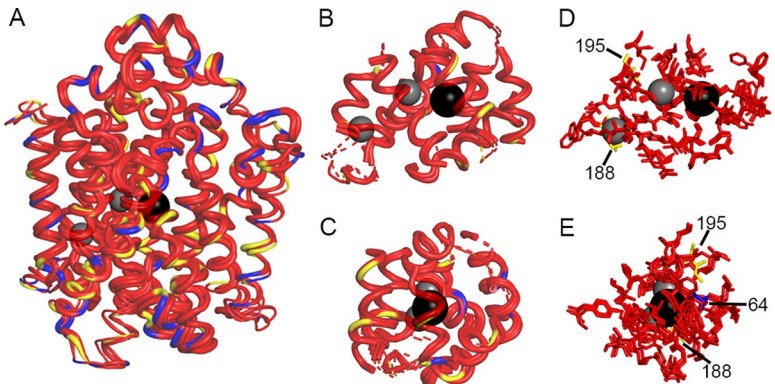

**Fig 8. Molecular models of human-, minke whale-, and zebrafish NIS.** (A) Overlap of hNIS, wNIS, and zNIS. The thickness of the alpha-helices and loops represent per-residue RMSD values calculated between the corresponding residues of hNIS, wNIS, and zNIS with hNIS as a reference. Thick alpha helices and loops signify low RMSD values (regions with high structural similarity). Color coding: red–fully conserved residues between hNIS, wNIS, and zNIS; yellow–substitution with a chemically similar residue in wNIS and zNIS; blue–substitution with non-similar residue; grey spheres–Na$^+$ ions, black sphere–I$^-$ ion. (B-C) Two different projections of protein areas within 10Å of Na$^+$ and within 15Å of I$^-$. Dashed lines indicate some or all of residues in the alpha helix are outside the 10/15Å cutoff. (D-E) Two different projections of residues from the ion coordination spheres (within 5Å) identified from MD simulations. The residue numbers of the non-conserved residues are also shown.

lower than wNIS and 3.94x lower than hNIS. Our uptake assay-derived $K_m$ value for hNIS is similar to a previous report of 9.0 μM, though the cell type we used is different from this study [69].

## Molecular modeling of NIS variants

The homology models of hNIS, wNIS and zNIS are overlapped in Fig 8A. The alpha-helices and loops of the three NIS species have different thickness, referring to the degree of structural similarities estimated from per-residue Root Mean Square Deviation (RMSD) values calculated between the corresponding residues of hNIS, wNIS and zNIS, using hNIS as a reference state. Protein areas with low RMSD values (i.e. high degree of structural similarity) are shown as thick alpha-helices and loops. The largest RMSD values (thinnest curves) were observed in mobile regions of the protein, such as the unstructured extra- and intracellular loops. TM12 (on the right side of the protein in Fig 8A) shows a high degree of structure deviation which likely arises from the lack of the constricting presence of TM13, which is missing in our model due to poor template availability. The color coding in Fig 8 reflects the degree of amino acid conservation in the NIS sequences of wNIS and zNIS, compared to hNIS: red depicts fully conserved residues, yellow–substitution with a chemically similar residue, and blue–substitution with a non-similar residue. The Na$^+$ and I$^-$ ions are shown as grey and black spheres, respectively. Fig 8B and 8C shows the protein surrounding the ions at the ion binding sites suggested by our Molecular Dynamics studies, within 15Å of I$^-$ and 10 Å of Na$^+$. The specific residues involved in ion coordination, identified from the MD simulations are shown in Fig 8D and 8E as sticks. S3 Fig shows the entire proteins (S3A Fig, cartoon representation) and the ion binding sites (S3B and S3C Fig, stick representations) for clearer depiction of the difference in the side chain orientation between hNIS (red), wNIS (blue), and zNIS (grey). The binding pockets of hNIS, wNIS and zNIS are highly conserved and structurally similar. The most significant differences between the three structures occur in more peripheral areas of the protein–the protein/lipid interface, and the intra- and extracellular loop regions of NIS (Fig 8A). Most of the residues in close proximity (~10–15 Å) to the bound Na$^+$ and I$^-$ ions are conserved or

substituted with chemically similar residues (Fig 8B and 8C). The non-conserved residues are positioned on alpha-helical turns facing away from the ions and may play a role in helical packing or impact indirectly ion binding as second and third coordination sphere residues. The residues in immediate proximity (~5 Å) to the bound $Na^+$ and $I^-$ ions are highly conserved, with the exception of conservative substitutions at positions 188 and 195 (V188I in wNIS and zNIS; V195I in zNIS) and a potentially impactful substitution at position 64 (S64A in wNIS and S64C in zNIS). The side chain orientations are similar for most binding residues in Fig 8D and 8E, reinforcing the structural similarity of the binding pockets of the three studied NIS proteins.

## Discussion

The mechanism by which NIS transports its various monovalent anionic substrates remains poorly understood. NIS does not transport every monovalent anion and it is still unclear what differentiates a substrate from an inhibitor or irrelevant anion [13, 83–87]. Here we compared the anion selectivity and inhibitor sensitivity of NIS proteins from diverse animal species and found, unexpectedly, that they differ not only in their ability to transport the NIS substrates $^{99m}TcO_4^-$ and $B^{18}F_4^-$, but also in their susceptibilities to inhibition of iodide transport by a wide range of monovalent anions. We performed more detailed analysis on the NIS proteins from three species living in natural habitats with differing iodide availabilities: zebrafish as a freshwater species, minke whale as a saltwater species, and human as a non-aquatic mammalian species. Interestingly, minke whale NIS (wNIS) was relatively resistant to perchlorate inhibition whereas zebrafish NIS (zNIS) was remarkably sensitive. In repeated experiments, wNIS proved to be more than 50-fold less sensitive than zNIS to perchlorate inhibition and human NIS (hNIS) showed intermediate sensitivity. This difference was recapitulated with every NIS substrate and inhibitor tested. Kinetic analysis of the transport properties of these proteins revealed that wNIS binds to iodide with the lowest affinity but transports it at the fastest rate. Molecular modeling suggests the putative ion binding pockets are well conserved, indicating the residues or regions responsible for the observed differences likely reside elsewhere in the protein.

It is possible that the differences observed may be the result of disparate protein expression levels. However, we are confident NIS expression was equalized and uptake data was normalized as much as current methods and protein peculiarities allowed, and that our results reflect fundamental differences in the structure and function of h-, w-, and zNIS. The human cell *in vitro* system used in this study may give hNIS an unfair advantage by providing a more natural environment for folding and maturation, but our ability to express wNIS and zNIS to similar levels in transfected cells (Fig 3A) and the equal level of iodide transport achieved by the three proteins (Fig 4E and 4F) suggests the human cell type did not impede non-human NIS protein expression or function. To date, no cellular factors have been reported which alter the ion transport function of NIS, so it is unlikely that wNIS or zNIS were restricted in human cells. One protein, leukemia-associated RhoA guanine exchange factor (LARG), is reported to interact with NIS, but does not affect ion transport and is considered to be a 'nonpump function' of NIS [88, 89].

Our findings are unexpected and suggest that the NIS proteins of different species may have diverged as a consequence of environmental pressures. Seawater has a much higher total chemical and ionic load than freshwater, with the total dry chemical weight of seawater being 288-333x greater than that of river water [32–34]. Iodide is transported from the diet to the bloodstream via NIS expressed in the intestines, a transport process which can presumably be inhibited by ingested non-iodide monovalent anions which are abundant in seawater but not

in fresh water (Table 1) [2]. This suggests that wNIS, in contrast to zNIS, evolved to efficiently transport iodide even in the presence of similar anions, which may explain its relatively higher anion transport promiscuity. Resistance to competitive inhibition appears to correlate with a high $V_{max}$ for iodide, which may extend to other competitive substrates (including perchlorate), allowing for effective transport of both iodide and the competitive substrate concurrently.

Why these NIS proteins handle competitive inhibition differently may be a product of evolutionary pressure and physiological multi-tasking. Aside from its role in concentrating iodide for thyroid hormonogenesis, NIS is a necessary component of the lactoperoxidase (LPO) antibacterial defense system in human airway epithelia since it transports $SCN^-$. $SCN^-$ is an ideal substrate for LPO, generating $OSCN^-$, which has antibacterial, antifungal, and antiviral activity [90]. $I^-$ can also be oxidized to $OI^-$ which behaves similarly to $OSCN^-$ [91]. The LPO system has also been identified in mammary, lacrimal, and salivary glands, and a similar peroxidase, DUOX2, has been reported in the ovary, fallopian tube, and uterus, suggesting NIS in these tissues may be involved in maintaining sterility [5, 92–94]. NIS is also expressed in the epithelium of the choroid plexus where it has been shown to transport $I^-$, $SCN^-$, $TcO_4^-$, and $Br^-$ out of the brain ventricle and into blood [95, 96]. It is therefore conceivable that NIS may play an important role in the exclusion of toxic monovalent anions from the central nervous system and that this may have provided an additional driver of NIS protein evolution in the face of differing species-specific environmental anion exposures. While there is currently no hard evidence to support this speculation, several studies have shown that excess iodide, bromine, and bromate can lead to oxidative stress, DNA damage, and apoptosis, especially in the brain [97–101].

At the molecular level, this study may provide some insight into the mechanism by which NIS is able to discriminate between small inorganic monovalent anions. Most tested substrates which lead to strong competitive inhibition of transport (Fig 6) are either tetrahedral or octahedral in shape. Such geometry allows for good multidirectional overlap with the residues lining the permeation pathways and binding pockets of NIS and may lead to stronger binding within the protein core, impacting both free energy of binding and transport kinetics. Several residues in hNIS where iodide transport-ablating mutations have been described are highly conserved across all examined species, indicating these are essential to NIS function (Fig 1 and S1 Fig, solid and open circles) [22]. Likewise, several NIS residues reported to play a role in stoichiometry and ion coordination do not vary between these proteins [52, 53]. However, since the proteins in this study clearly respond to the tested substrates differently, this indicates the previously described residues are not the sole sites of anion selectivity and ion coordination (Fig 1 and S1 Fig, bold red residues). Of thirteen charged residues located on the extracellular loops of hNIS which were reported to significantly reduce iodide uptake when individually mutated to alanine, zNIS lacks four: R82L, D163N, H226G, and R239Q (Fig 1, red triangles) [71]. zNIS has four additional positively charged residues (Fig 1, open diamonds) and wNIS has three additional negatively charged residues (Fig 1, blue diamonds) on the extracellular face. This variation in outward-facing charged residues may be a potential source of substrate transport modulation. Charged residues in the extracellular loops were recently implicated in inhibitor selectivity in human SGLT2, a sodium glucose symporter related to NIS [102]. Studies are ongoing to determine the impact of extra- and intracellular residue differences between these NIS proteins.

To investigate a possible structural source for the observed data, we generated a molecular model of hNIS based on two proteins, vSGLT and Mhp1, which share the same LeuT-fold architecture. wNIS and zNIS were then mapped onto the hNIS protein model and their side chains were optimized with ROSETTA MP (Fig 8). The first hypothesis was that the selectivity

differences presented in Figs 4–7 arise due to changes in residues in the ion binding pocket directly involved in ion coordination. In this model, these are residues S62, S64-S69, V71, Q72, Q94, Y144, V148, Y178, M184-D191, Q194, V195, M198, W255, L256, Y259, N262, Q263, Q265, L289, I292, S349, G350, S353, T354, T357, S358, F417, and M420. Thorough validation of this hNIS model and identification of the residues involved in $Na^+$ and $I^-$ binding from Molecular Dynamics simulations and functional mutagenesis is under revision elsewhere. Mapping of the residues in direct contact with the transported ions revealed no significant differences in the sampled models in terms of residue identity and side chain positions between the three species of NIS protein (Fig 8B–8E). Several of these residues have been previously reported as crucial for NIS activity [53, 85, 86]. Within the hNIS, wNIS, and zNIS subset, the differing residues in the binding areas are at positions 64, 188, and 195, where only the substitution at position 64 (S64C in wNIS and S64A in zNIS) leads to change in the chemical properties of the side chain. Residue S64 (blue in Fig 8D and 8E) coordinates one of the $Na^+$ ions (Na1) via its carbonyl oxygen and its side chain points away from the bound ions. A S64A mutation has been assessed previously and showed no significant impact on $I^-$ uptake, implying that the -OH group at this position is not critical for binding [70].

This study represents the first report, to our knowledge, of NIS variants with substantive differences in substrate affinity and inhibitor sensitivity that are still capable of competently transporting iodide. Previously reported rat NIS-G93X mutants exhibited altered substrate stoichiometry and reduced perchlorate affinity and sensitivity. However, these mutants lost substantial affinity for iodide compared to WT NIS [52]. Here we identified NIS proteins which responded differently to competitive substrates and inhibitors while maintaining efficient iodide transport, though there seemed to be a tradeoff between affinity and inhibition resistance. Our data indicate a wide degree of variability both within one NIS protein and across different species of NIS regarding iodide uptake inhibition by monovalent anions. Almost all affinity and transport values reported in the literature were determined with rat NIS. The NIS field has long held rat NIS-derived values to be representative of hNIS, but our data suggest this may not be the case. The assumption that each species of NIS protein will respond identically to a certain substrate or inhibitor is no longer reasonable. Additionally, our study suggests that it might be possible to engineer NIS proteins with preferentially increased uptake of particular substrates or resistance to inhibitors, which may in turn enhance the utility of NIS in many biotechnological and clinical applications. Further studies are ongoing to further elucidate the basis of the observed differences in monovalent anion substrate selectivity and sensitivity to perchlorate inhibition between the NIS proteins of distantly related species.

## Supporting information

**S1 Fig. Amino acid sequence alignment of the NIS proteins examined in this study.** *H. sapiens* (human), *P. anubis* (olive baboon), *B. acutorostrata scammoni* (minke whale), *T. truncatus* (bottle-nosed dolphin), *C. lupus familiaris* (dog), *S. scrofa* (pig), *R. norvegicus* (rat), *M. musculus* (mouse), *H. leucocephalus* (bald eagle), *P. sinensis* (Chinese soft-shell turtle), *X. laevis* (African clawed frog), *D. rerio* (zebrafish), and *C. harengus* (Atlantic herring). Cyan highlighting indicates absolute conservation to human NIS. Yellow indicates similar residue to human NIS. Underline indicates putative transmembrane domain in human NIS, only TM1-12 are indicated. Closed circles indicate site of a mutation known to cause a transport defect in humans [22]. Open circles indicate site of a mutation known to cause membrane trafficking defect in humans [22]. Black triangles indicate a charged residue where mutation to alanine significantly reduces iodide uptake in human NIS [71]. Red triangles indicate a charged residue where

mutation to alanine significantly reduces iodide uptake in human NIS and this residue is not charged in zebrafish NIS [71]. Bold red lettering indicates residue reported to be involved in stoichiometry control and translocation dynamics [52, 53]. Species are ordered in ascending evolutionary proximity to humans as determined by TimeTree (pig, mouse, rat, and dog diverged equidistantly) [73]. Numbering follows human NIS.
(TIF)

**S2 Fig. Competitive substrate inhibition assays in HeLa cells transduced with six additional species of NIS protein.** (A) HA-African clawed frog NIS, (B) HA-pig NIS, (C) HA-mouse NIS, (D) HA-rat NIS, (E) HA-dog NIS, or (F) HA-olive baboon NIS. Data shown as the percentage of maximum $^{125}I^-$ uptake activity (0 μM compound) maintained in the presence of increasing concentrations of substrate or inhibitor. Circular markers indicate naturally occurring anions. Triangular markers indicate anions which may occur naturally at low levels or are generated inside the organism. Square markers indicate anions not found naturally. Values are averages of duplicate assays with standard deviation.
(TIF)

**S3 Fig. Additional molecular models of human-, whale-, and zebrafish NIS. (**A) Overlap of hNIS (red), wNIS (blue), and zNIS (grey). The $Na^+$ and $I^-$ are presented as yellow and orange spheres, respectively. Color coding follows the colors used for Figs 3–5 and 7. (B-C) Two different projections of residues from the ion coordination spheres (within 5Å) identified from our MD simulations. The same projections were used in Fig 8D and 8E.
(TIF)

## Acknowledgments

We thank Yasuhiro Ikeda for providing the lentiviral plasmid. We are grateful to Patrycja Lech for her pioneering work on NIS in the Russell lab and developing many of the experimental procedures used in this study. We also thank Kah-Whye Peng, Toshie Sakuma, Lukkana Suksanpaisan, Ryan Johnson, Arun Ammayappan, and Michael Romero for their enlightening input during discussions. We thank Robert Kangas for his detailed proofreading.

## Author Contributions

**Conceptualization:** Susanna C. Concilio, Hristina R. Zhekova, Sergei Y. Noskov.

**Data curation:** Susanna C. Concilio, Hristina R. Zhekova.

**Formal analysis:** Susanna C. Concilio, Hristina R. Zhekova.

**Funding acquisition:** Hristina R. Zhekova, Sergei Y. Noskov, Stephen J. Russell.

**Investigation:** Susanna C. Concilio, Hristina R. Zhekova.

**Methodology:** Susanna C. Concilio, Hristina R. Zhekova, Sergei Y. Noskov.

**Project administration:** Sergei Y. Noskov, Stephen J. Russell.

**Resources:** Sergei Y. Noskov, Stephen J. Russell.

**Software:** Susanna C. Concilio, Hristina R. Zhekova, Sergei Y. Noskov.

**Supervision:** Sergei Y. Noskov, Stephen J. Russell.

**Validation:** Susanna C. Concilio, Hristina R. Zhekova.

**Visualization:** Susanna C. Concilio, Hristina R. Zhekova.

Writing – **original draft:** Susanna C. Concilio, Hristina R. Zhekova.

Writing – **review & editing:** Susanna C. Concilio, Hristina R. Zhekova, Sergei Y. Noskov, Stephen J. Russell.

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
