## [Decision Letter · Decision Letter 0]

24 Jan 2020

PONE-D-19-35941

Inter-species variation in monovalent anion substrate selectivity and inhibitor sensitivity in the sodium iodide symporter (NIS)

PLOS ONE

Dear Dr. Russell,

Thank you for submitting your manuscript to PLOS ONE. After careful consideration, we feel that it has merit but does not fully meet PLOS ONE’s publication criteria as it currently stands. Therefore, we invite you to submit a revised version of the manuscript that addresses all the points raised by the reviewer. Reducing the text in selected places would improve the manuscript.

We would appreciate receiving your revised manuscript by Mar 09 2020 11:59PM. To enhance the reproducibility of your results, we recommend that if applicable you deposit your laboratory protocols in protocols.io, where a protocol can be assigned its own identifier (DOI) such that it can be cited independently in the future. For instructions see: http://journals.plos.org/plosone/s/submission-guidelines#loc-laboratory-protocols

We look forward to receiving your revised manuscript.

Kind regards,

Claudio M. Soares, Ph.D

Academic Editor

PLOS ONE

Journal Requirements:

"I have read the journal's policy and the authors of this manuscript have the following competing interests: S.J.R. is a cofounder and holds equity in Imanis Life Sciences, LLC. S.C.C., S.Y.N, and H.R.Z. declare no competing interests."

Reviewers' comments:

Reviewer's Responses to Questions

**Comments to the Author**

1. Is the manuscript technically sound, and do the data support the conclusions?

Reviewer #1: Yes

2. Has the statistical analysis been performed appropriately and rigorously? 

Reviewer #1: Yes

3. Have the authors made all data underlying the findings in their manuscript fully available?

Reviewer #1: Yes

4. Is the manuscript presented in an intelligible fashion and written in standard English?

Reviewer #1: Yes

5. Review Comments to the Author

Reviewer #1: The current manuscript by Dr Concilio and colleagues examines the functional parameters of several species’ NIS proteins, and models structural components of them.

This is an interesting study that I think merits publication, albeit the data are slightly ‘flat’ in terms of the lessons we learn about iodide transport (as well as of related compounds) in the various species studied. However, this is an extremely thorough body of work, considering multiple compounds in multiple species. This is the kind of manuscript that will provide source data for scientists working in the field to pore over and use almost as a reference text, and hence I believe it will be well cited.

One philosophical point: the relationship between substrate transport efficiency and iodide content in the environment may be less likely to reflect the comparative protein structure of NIS, and any evolutionary adaptations in terms of amino acid sequence, and more likely to reflect NIS expression levels and subcellular localisation. This point should be addressed.

In terms of specific criticisms of the work, there aren’t many as the findings are straightforward and the systems used are well characterised. I could quibble about the choice of cell lines used (HeLa and HEKs), both of which are human and non-thyroidal. It is thus possible that the zebrafish NIS gene is different in iodide transport to the human gene simply because it is being tested in a human and not zebrafish setting. Furthermore, human proteins known to bind NIS and alter its function may not interact with e.g. zebrafish NIS in the same way. Although it is not feasible to repeat the experiments in zebrafish and minke whale cell lines, the authors should comment.

Small points:

1. I'm not sure the manuscript as submitted matches the standard format of PLOS One papers. It may be my naivety but I was not sure that Figure legends should be embedded in the Results section?

2. The paper is too long, and a lot of the experimental detail can be condensed, and the Discussion cut by 50%.

6. PLOS authors have the option to publish the peer review history of their article (what does this mean?). If published, this will include your full peer review and any attached files.

Reviewer #1: No

---

## [Author Response · Author response to Decision Letter 0]

28 Jan 2020

We greatly appreciate the timely manner in which this review was conducted. We thank the editor-in-chief and Dr. Soares for their role in this process and assistance with the submission process. We also thank the reviewer for his/her detailed reading of our manuscript and his/her assessment of the strengths and weaknesses, as well as insightful suggestions.

We have edited the manuscript to adhere to PLOS One’s style and requirements. We have run our figures through the Preflight Analysis and Conversion Engine (PACE) and they meet PLOS One’s publication standards. We have also shortened the Materials and Methods, Results, and Discussion sections as much as possible, per Reviewer 1’s recommendation. 

In regards to Reviewer 1’s specific comments, we have addressed the two major concerns with additional text in the manuscript. 

1) Reviewer 1: “[T]he relationship between substrate transport efficiency and iodide content in the environment may be less likely to reflect the comparative protein structure of NIS, and any evolutionary adaptations in terms of amino acid sequence, and more likely to reflect NIS expression levels and subcellular localisation. This point should be addressed.”

To expand upon our addition to the discussion about this point, we believe our results are indicative of fundamental attributes of the NIS proteins examined in this study as we made every attempt to equalize cell surface expression both during the experiment via the creation of cell lines with near-identical expression and after the experiment by normalization of uptake values to expression levels as determined by flow cytometry of transfected and transduced cell lines. Any differences in substrate transport observed should be the result of inherent protein function rather than differences in protein maturation, cell surface expression, or subcellular localization. 

2) Reviewer 1: “Furthermore, human proteins known to bind NIS and alter its function may not interact with e.g. zebrafish NIS in the same way. Although it is not feasible to repeat the experiments in zebrafish and minke whale cell lines, the authors should comment.” 

We cannot deny the possibility that minke whale NIS and zebrafish NIS may have altered function due to being tested in a human cell environment. However, two points support our results and conclusions despite the potential human NIS bias: 1) As mentioned in the updated discussion, to date, there are no cellular factors known to bind or interact with NIS which affect the ion transport function. Two papers have reported NIS interacts with leukemia-associated RhoA guanine exchange factor (LARG) (DOI:10.1158/0008-5472.CAN-12-0516 and DOI: 10.1158/0008-5472.CAN-18-1954). However, this interaction was not shown to have any affect upon NIS transport activity and is considered a nonpump function of NIS. 2) Figure 2E and 2F indicates that human-, minke whale-, and zebrafish NIS are capable of transporting iodide to the same degree, suggesting there is no restriction of minke whale- or zebrafish NIS function in human cells. We also acknowledge that this manuscript does not provide experimental evidence of a mechanism for the observed differences, but we are confident that the advances we offer will be the source of much discussion and future work in the NIS field.

---

## [Decision Letter · Decision Letter 1]

30 Jan 2020

Inter-species variation in monovalent anion substrate selectivity and inhibitor sensitivity in the sodium iodide symporter (NIS)

PONE-D-19-35941R1

Dear Dr. Russell,

We are pleased to inform you that your manuscript has been judged scientifically suitable for publication and will be formally accepted for publication once it complies with all outstanding technical requirements.

With kind regards,

Claudio M. Soares, Ph.D

Academic Editor

PLOS ONE

Additional Editor Comments (optional):

Reviewers' comments:

Reviewer's Responses to Questions

**Comments to the Author**

1. If the authors have adequately addressed your comments raised in a previous round of review and you feel that this manuscript is now acceptable for publication, you may indicate that here to bypass the “Comments to the Author” section, enter your conflict of interest statement in the “Confidential to Editor” section, and submit your "Accept" recommendation.

Reviewer #1: All comments have been addressed

2. Is the manuscript technically sound, and do the data support the conclusions?

Reviewer #1: Yes

3. Has the statistical analysis been performed appropriately and rigorously? 

Reviewer #1: Yes

4. Have the authors made all data underlying the findings in their manuscript fully available?

Reviewer #1: Yes

5. Is the manuscript presented in an intelligible fashion and written in standard English?

Reviewer #1: Yes

6. Review Comments to the Author

Reviewer #1: The manuscript has been shortened, and some points made clearer/discussed more comprehensively.

Thank you for addressing my points.

7. PLOS authors have the option to publish the peer review history of their article (what does this mean?). If published, this will include your full peer review and any attached files.

Reviewer #1: No

---

## [Editor Report · Acceptance letter]

13 Feb 2020

PONE-D-19-35941R1 

Inter-species variation in monovalent anion substrate selectivity and inhibitor sensitivity in the sodium iodide symporter (NIS) 

Dear Dr. Russell:

I am pleased to inform you that your manuscript has been deemed suitable for publication in PLOS ONE. Congratulations! Your manuscript is now with our production department. 

With kind regards,

on behalf of

Dr. Claudio M. Soares 

Academic Editor

PLOS ONE